# Management of Recurrent Retroperitoneal Sarcoma

Joshua S. Jolissaint [1] , Chandrajit P. Raut [1,2] and Mark Fairweather [1,2,*]

1    Department of Surgery, Brigham and Women's Hospital, Harvard Medical School, Boston, MA 02115, USA
2    Center for Sarcoma and Bone Oncology, Dana-Farber Cancer Institute, Harvard Medical School, Boston, MA 02115, USA
*    Correspondence: mfairweather@bwh.harvard.edu; Tel.: +1-(617)-842-4612; Fax: +1-(617)-582-6177

**Abstract:** Recurrence after resection of retroperitoneal sarcoma is common and varies by histological subtype. Pattern of recurrence is similarly affected by histology (e.g., well-differentiated liposarcoma is more likely to recur locoregionally, whereas leiomyosarcoma is more likely to develop distant metastases). Radiotherapy may provide effective locoregional control in limited circumstances and the data on the impact of chemotherapy are scant. Surgery for locally recurrent disease is associated with the greatest survival benefit; however, data are retrospective and from a highly selected subgroup of patients. Limited retrospective data have also suggested a survival association with the resection of limited distant metastases. Given the complexity of these patients, multidisciplinary evaluation at a high-volume sarcoma center is critical.

**Keywords:** sarcoma; retroperitoneal; recurrent; surgery

## 1. Introduction

Retroperitoneal sarcomas (RPS) are a rare and heterogenous group of tumors, which includes well-differentiated (WD-) or dedifferentiated liposarcoma (DDLPS), leiomyosarcoma (LMS), solitary fibrous tumor (SFT), malignant peripheral nerve sheath tumor (MPNST) and undifferentiated pleomorphic sarcoma (UPS) [1]. Although surgical resection at the time of initial presentation affords the best opportunity for a long-term cure, recurrence is common, ranging between 20% to 55% and varying considerably by histological subtype and site of recurrence; local recurrence has been reported as high as 80% for primary DDLPS [2–6]. As an example, in a single institution series of 675 patients from Memorial Sloan Kettering, Tan et al. noted that both SFT and LMS have a comparatively low incidence of LR (8% and 16% at 3 years) but a higher incidence of distant recurrence (41% and 58% at 10 years) [3]. In contrast, WDLPS had a 60% incidence of LR over 15 years and a low risk of distant metastasis at 8% over 10 years. DDLPS also recurs locally, but does so before WDLPS, with 58% of patients experiencing LR at 5 years (and only rising 62% by 15 years). However, in contrast to WDLPS, DDLPS has a cumulative incidence function (CIF) of distant metastasis of 28%. Similarly, in a multi-institutional series of 1006 patients by Gronchi et al., local and distant recurrence were analyzed using a Cox Proportional Hazards model with SFT as the referent [6]. UPS (HR 3.4) and WDLPS (2.25) were more likely to recur locally, whereas LMS was more likely to recur at a distant site (HR 2.90). Additional factors that have been associated with the risk of local recurrence include grade, size, contiguous organ resection, and R1 or R2 resection [2,3,7].

Primary RPS often abuts adjacent organs and an extended en bloc resection to achieve a macroscopically complete resection may require multivisceral resection and attention to vascular control (Figures 1 and 2) [8]. While this approach has demonstrated improved local disease control and survival compared to historical data with acceptable morbidity, it is often technically difficult, and patients may benefit from referral to high volume hospitals (HVHs) [6,9,10]. While the annual case volume that constitutes "high volume" is unclear, an analysis of 6950 patients in the National Cancer Database (NCDB) by Keung et al.

demonstrated that patients treated at HVHs (defined in their manuscript as >10 cases per year) for primary RPS have lower 30-day readmission rates, 30- and 90-day mortality, as well as longer survival [11,12]. Similarly, Bonvalot et al. analyzed 382 patients with primary RPS and found that treatment at centers where ≥30 patients undergo surgery for RPS per year was associated with improved abdominal RFS [13]. Finally, Toulmonde et al. performed a retrospective analysis of 568 patients with primary RPS and found that treatment at a center specializing in sarcoma surgery was associated with a lower rate of LR (HR, 0.5 95% CI 0.4–0.7, *p* < 0.001) and abdominal sarcomatosis (HR 0.5, 95% CI 0.3–0.9, *p* = 0.02) [14]. As such, particularly for recurrent disease, which is often more nuanced and involves the complexities associated with potential reoperation, the Transatlantic Australasian RPS Working Group (TARPSWG) recommends referral to a center that specializes in RPS and supports the infrastructure to care for these patients in a multidisciplinary setting, involving surgeons, medical oncologists, radiation oncologists, pathologists, and radiologists [1,15].

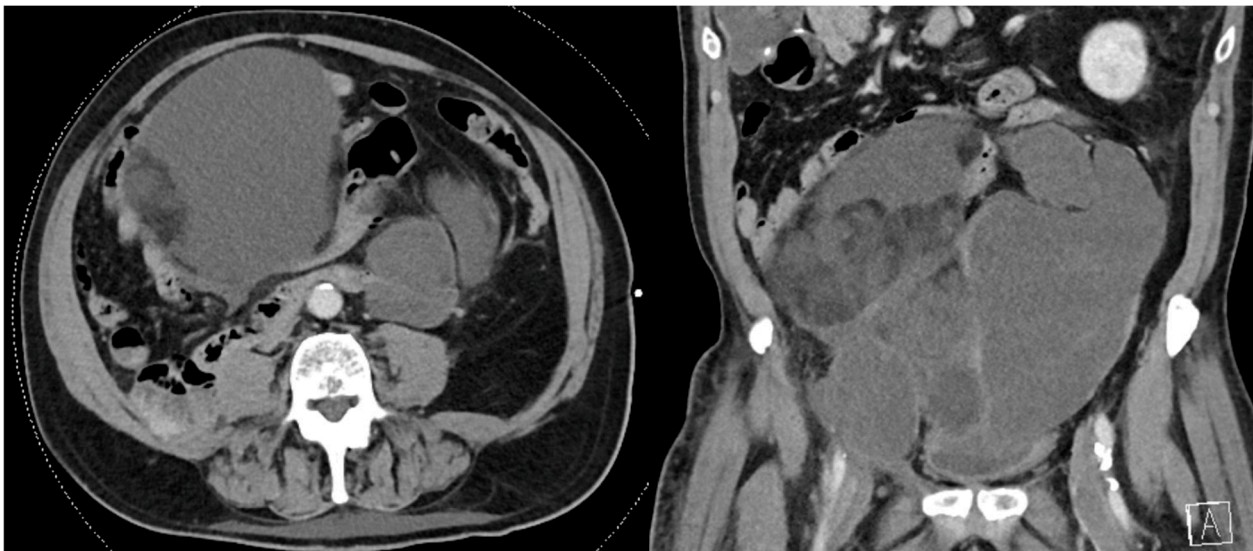

**Figure 1.** Recurrent multifocal de-differentiated liposarcoma requiring en bloc small bowel resection, sigmoid colectomy, and end colostomy. The index operation required en bloc right colectomy, right nephroureterectomy, right adrenalectomy, partial right psoas muscle resection, and right spermatic cord resection.

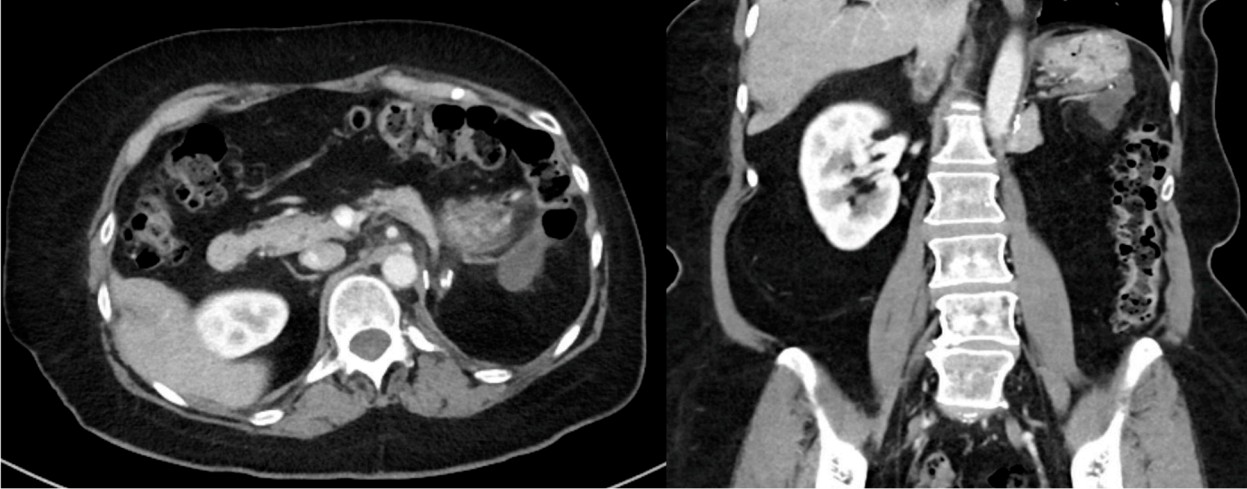

**Figure 2.** Recurrent well-differentiated liposarcoma requiring en bloc gastric wedge resection and partial colectomy. The index operation required en bloc left nephroureterectomy, left adrenalectomy, distal pancreatectomy, and splenectomy.

Similarly, rates of discordance between initial pathological review and second opinion by a pathologist specializing in sarcomas range from 24–40%, with up to 16% being clinically significant [16,17]. As such, we recommend centralized review of histopathology slides from the primary tumor and any recurrent biopsies by a pathologist with expertise in sarcomas as this may guide or change management. While some tumors have characteristic features, some histologic subtypes of RPS have histological features (specifically tumor cell morphology and growth patterns) that often overlap with other tumors. Immuno-histochemistry (IHC) may be used to help differentiate tumors of uncertain lineage, and may include protein-directed antibodies, or markers such as SMA, desmin, myogenin, CD34, S100 protein, MDM2, CDK4, STAT6, ALK, CD99, H3K27me3, NKX2.2, TLE1, SOX10, HMB-45, melan-A, cyclin D1, cytokeratin, and EMA [18]. Similarly, karyotype testing, fluorescence in situ hybridization (FISH) and next-generation genomic sequencing (NGS) may be used in the case of uncertain histologic and IHC findings, such as MDM2 amplification in WDLS and DDLS, and NAB2-STAT6 fusion for SFT [18,19]. Core needle biopsy of the recurrent tumor may be a useful adjunct in cases of uncertain diagnosis or when imaging is not confirmatory (e.g., in the case of liposarcoma), or if there is concern for a second malignancy [15,20].

## 2. Imaging

Recurrence is often diagnosed on surveillance imaging and frequently predates symptoms. Consequently, computed tomography (CT) is often the initial diagnostic modality and high quality, contrast-enhanced imaging can indeed help delineate elements of tissue composition. In the case of recurrent disease, cross-sectional imaging of the chest, abdomen, and pelvis should be included to assess for both metastatic disease and the extent of local tumor progression [15,20]. Magnetic resonance imaging (MRI) may be useful in cases of equivocal imaging findings on CT, pelvic disease, involvement of the muscle, bone, spine, nerve roots, or major vessels, or in the case of planned preoperative radiotherapy (RT) to best assess local tumor extent and tissue edema. $^{18}$F-fluorodeoxyglucose positron emission tomography ($^{18}$F-FDG PET) may be considered in order to better delineate the presence or absence of distant metastases; however, use of PET is not routine. Nevertheless, a recent study on 58 patients with DDLPS and LMS by Subramaniam et al. has also demonstrated an association between maximum standardized uptake value (SUVmax) with pathologic grade ($r_s = 0.4$, $p = 0.003$), as well as worse recurrence-free survival (RFS) ($p = 0.003$), and overall survival (OS) ($p = 0.003$), which may have future implications in the decision-making surrounding treatment of recurrent disease [21]. Further investigation and validation are necessary.

## 3. Chemotherapy

Chemotherapy has limited efficacy in the management of recurrent RPS and its use is a controversial topic, with data extrapolated from the locally advanced/metastatic setting and including varying tumor histology, grades, and anatomic locations [22–25]. In the neoadjuvant setting, the multicenter ISG-STS 1001 trial, which also included patients with previously resected sarcoma, found no benefit to a histology-tailored chemotherapy approach for "high risk" tumors (high grade, ≥5 cm, deep to investing fascia, myxoid liposarcoma, leiomyosarcoma, synovial sarcoma, MPNST, and UPS) over a standard regimen of epirubicin 60 mg/m$^2$ per day plus ifosfamide 3 g/m$^2$ per day [26]. However, in a multicenter, randomized, phase 3 trial by Issels et al. and the European Organization for Research and Treatment of Cancer (EORTC), patients with high-risk soft-tissue sarcoma (STS) (grade 2 or 3, ≥5 cm, deep to investing fascia), including recurrent tumors, were randomized to receive neoadjuvant therapy with four cycles of etoposide (125 mg/m$^2$ on day 1 and 4), ifosfamide (1500 mg/m$^2$ on days 1–4), and doxorubicin (50 mg/m$^2$ on day 1) (EIA) alone or in combination with regional hyperthermia [27]. In this study, 56% of patients had non-extremity STS (abdominal or retroperitoneal) and 11% were recurrent tumors. Patients receiving EIA plus regional hyperthermia had improved local progression

(HR 0.58) and DFS (HR 0.70) as well as OS (HR 0.66) based on a pre-specified per-protocol analysis. While this would argue in favor of using this combination approach, there was no control arm evaluating outcomes without chemotherapy; this chemotherapy regimen would not be considered standard of care at many centers, and regional hyperthermia is only available at a limited number of mostly Western European centers. Although limited to primary DDLPS and LMS, the STRASS 2 Trial (EORTC 1809, NCT04031677) is currently enrolling patients to determine whether neoadjuvant chemotherapy improves outcomes, as measured by DFS, OS, and RFS.

Regarding the adjuvant setting, the multicenter EORTC 62931 trial evaluated patients with grade II-III soft-tissue sarcoma (STS) at any site, including recurrent tumors, and found no benefit to a regimen of five cycles of doxorubicin (75 mg/m$^2$), ifosfamide (5 g/m$^2$) and lenogastrim (3 μg/kg) [28]. Recurrence-free survival (RFS) was 6.5 years in the no chemotherapy/control arm, compared to 7.6 years in the chemotherapy arm. Five-year RFS was 52.9% vs. 54.9% for the two arms, respectively. Notably, the cumulative incidence of distant metastases was 35% in both groups. Similarly, median overall survival (OS) was 12.4 years for the no chemotherapy arm and 11.2 years for the chemotherapy arm with a 5-year survival of 67.8% and 66.5%, respectively. It is important to note that only 21% of this study cohort had a central disease burden, with the remainder being either extremity or limb girdle. However, an analysis of this data using the Sarculator prognostic nomogram did find a benefit to adjuvant therapy for "high risk" patients, or those with a low predicted OS (pr-OS $\leq$ 51%) [29]. In this analysis, adjuvant therapy reduced the risk of both recurrence (HR 0.46) and death (HR 0.46) by approximately half.

## 4. Radiation

There are limited data on the use of radiation for recurrent RPS. The multicenter, phase 3 EORTC-62092 STRASS trial randomized 266 patients with primary RPS to receive either 50.4 Gy of preoperative radiation (28 fractions of 1.8 Gy) or upfront resection and found no benefit to preoperative radiation [30]. A post hoc sensitivity analysis of liposarcoma histology suggested an improvement in abdominal recurrence-free survival; however, this trial only included primary RPS and excluded patients with prior therapy or resection. Patients with recurrent RPS were not eligible for the STRASS trial, and there have been no randomized clinical trials (RCTs) in the recurrent setting and most experiences are limited to case series [31].

In a 2007 series on 85 patients treated with radiotherapy (RT) for RP and deep truncal sarcoma (7–73 Gy, median 56.4 Gy), Feng et al. reported 2- and 5-year local control and OS of 66% and 51%, and 70% and 34%, respectively [32]. In this study, 35% of patients had LMS, 20% LPS, 17% UPS (referred to as malignant fibrous histiocytoma in this series), and 9% MPNS. Although radiation dose was associated with survival on univariate analysis, when removing patients whose RT was discontinued prematurely (after 7 and 33 Gy), this factor was no longer statistically significant ($p$ = 0.4). In 2009, Serizawa et al. published a series on 24 patients treated with carbon ion radiotherapy (CIRT) for unresectable RPS and demonstrated a favorable survival with 2- and 5-year OS of 75% and 50%, as well as local control rates of 77% and 69%, respectively. Most recently, Yang et al. described their experience using CT-guided $^{125}$I seed implantation (mean of 70.87 seeds (range 10–210) for 23 unresectable RPS with an 87% local control rate, median OS of 21.6 months, with significantly improved pain ratings based on a visual analog scale (VAS) after treatment [33].

## 5. Resection

When technically feasible and in a medically fit patient, reoperation and repeat resection may yield favorable survival outcomes compared to other modalities; however, this must be weighed against the morbidity and mortality associated with repeat resection [34]. Moreover, this has never been compared in the clinical trial setting, and retrospective data are inherently fraught with selection bias. In the case of local recurrence, the technical principles are analogous to those of primary RPS and should be performed with curative

intent, involving resection of all involved organs en bloc; however, these operations are often challenging due to adhesions and distorted anatomy (Table 1).

**Table 1.** Brief Technical Principles of Retroperitoneal Sarcoma Resection.

| |
| --- |
| Incision should maximize exposure and minimize the risk of tumor capsule disruption. |
| A midline laparotomy has the benefit of adequate access to the retroperitoneum, as well as future access in the case of local recurrence. |
| Chevron, Makuuchi, or thoracoabdominal incisions may provide superior exposure for upper quadrant sarcomas. |
| A transverse flank or modified Gibson incision may provide superior exposure of the superior pelvis and iliac vessels, particularly for extraperitoneal pelvic sarcomas. |
| While access to the retroperitoneum may involve mobilization of the colon, a macroscopically complete resection often necessitates en bloc colectomy and resection of any additional involved organs. |
| Preoperative planning and sterile preparation should anticipate the possibility of an end or diverting ostomy. |
| Right-sided tumors may require nephroureterectomy with Kocherization of the duodenum and head of the pancreas and include ipsilateral colectomy, adrenalectomy, and psoas/psoas fascia resection. |
| Left-sided tumors may require nephroureterectomy, as well as distal pancreatectomy and splenectomy and include ipsilateral colectomy, adrenalectomy, and psoas/psoas fascia resection. |
| Tumors arising from the lower 1/3 of the inferior vena cava (IVC) may require a full Cattell-Brasch maneuver for exposure. |
| Tumors arising from the middle 1/3 of the IVC may require hepatic resection of uninvolved tissue for an R0 resection. |

Adapted with permission from [34].

In a retrospective series multi-institutional series by TARPSWG, MacNeill et al., 40.5% (408/1007) of patients developed recurrent disease after resection of primary RPS during the follow-up period [35]. Of those who recurred, the initial site of recurrence was local recurrence (LR) in 53.7% (219/408), distant recurrence (DR) in 35.8% (146/408) and simultaneous local and distant recurrence in 10.5% (43/408). The median OS after LR was 33-months with a 5-year OS of 29%, whereas the median OS after DR was 25 months with a 5-year OS of 20%. Repeat resection demonstrated a statistically significant association with improved survival (HR for no resection, 3.96 95% CI 2.32–6.76, $p$ <0.001) but was not significant for distant recurrence (HR 1.62, 95% CI 0.97–2.74, $p$ = 0.0668). Patients who underwent repeat resection had a median OS of 49 months (vs. 20 months for those who did not).

Raut et al. analyzed 684 patients from 22 centers who underwent surgery for first relapse locally recurrent RPS (RPS-LR1) [36]. Among this cohort, 6-year DFS and OS were 19.2% and 54.1%, respectively. The median disease-free interval (DFI) after the second surgery was 19 months, which was a second local recurrence in 58% of cases. Both recurrence and OS were influenced by histology. Notably, the 6-year crude cumulative incidence (CCI) of local recurrence ranged from 60.2–70.9% for WD- and DDLS and 71.4% for SFT, with lower rates of distant recurrence. Conversely, the 6-year CCI of distant recurrent was 50% for MPNST and 36.3% for LMS (with a 44.0% CCI of local recurrence).

The same group assessed 567 patients from 22 centers who underwent macroscopically complete resection of a first local recurrence (LR), specifically looking at the outcomes of 400 patients who developed a second local recurrence, 200 of whom would undergo a repeat surgical resection [37]. The authors showed that pattern of failure significantly influenced survival; 5-year OS was 45.6% for LR, 25.5% for distant metastases (DM) and 0% for simultaneous LR and DM. This pattern was heavily influenced by resectability, as the median OS was 77 months for resected patients (IQR 34–96 months) compared to 18 months in those who were ineligible for resection (IQR 7–40 months). Interestingly, on multivariate analysis, histological type and tumor grade were not significant predictors of OS. The only factors associated with improved survival on multivariate analysis were time-to-recurrence (HR 0.44) and resection of a second LR (ref. yes, HR 3.25). Similarly, in a

series of 55 patients by Tropea et al., histological type was not associated with OS even at the time of resection for first LR; however, this may have been limited by the smaller cohort size [38].

In a smaller single-institution series of 95 primary RPS resections by Yang et al., 50 patients underwent a second repeat resection; repeat resection with 1-year, non-liposarcoma histology, higher histological grade, and gross residual disease were associated with worse OS [39]. However, when analyzing the 26 patients who underwent a third resection, histological grade did not significantly impact OS ($p = 0.058$).

While most reports detailed herein describe repeat/salvage surgery for local recurrence, surgery for distant recurrence may have good survival outcomes in carefully selected patients. In a retrospective series of 172 patients with RP LMS, which as previously described has a higher propensity for distant metastases as opposed to local recurrence, Ikoma et al. showed that patients who underwent salvage surgery had a 2.3-year survival benefit (5.6 vs. 3.3 years), and site of recurrence was not associated with OS (HR 1.45, 95% CI 0.880–2.71, $p = 0.134$) [40]. In this series, 18.6% experienced local recurrence, 41.9% developed distant metastases, and 5.8% had simultaneous local recurrence and distant metastases. Lung was the most common site of distant metastases and 32/72 (44.4%) of patients with distant metastases underwent salvage resection.

## 6. Patient Selection

The decision to proceed with further treatment at the time of recurrence is multifactorial. Frailty, comorbidities, and patient performance status should always factor into the treatment algorithm, as well as baseline organ function as it relates to systemic chemotherapy administration or the need for multivisceral repeat resections (e.g., nephrectomy as part of an en bloc resection). Further, the disease-free interval is a surrogate for tumor biology, correlates with OS, and should be used as a factor when considering repeat resection [34,35,39]. Multifocal disease, particularly simultaneous DR and LR, is associated with worse OS and should likely only be undertaken for palliation without a demonstrable survival benefit [34,41]. As detailed below, repeat resection at the time of first recurrence has favorable outcomes and some institutions have adopted an aggressive resection policy. Lv et al. offer their institutional algorithm and generally offer resection in cases of macroscopically resectable locally recurrent tumors after taking into consideration the necessity of bowel resection and concomitant short gut syndrome, and the possibility of permanent colostomy or urinary diversion [42]. Resection may be offered for cases of distant metastases when the patient can be rendered disease-free, or in the case of unresectable tumors in the case of symptom palliation, such as bleeding or obstruction. Nomograms to guide the decision to repeat resection have been developed. Raut et al. created a nomogram to predict both 6-year DFS and OS, notably, multifocality, tumor grade, macroscopic resection, histology, adjuvant chemo-/radiotherapy, and multivisceral resection at the time of the first resection are all important considerations and may factor into outcomes for these patients [36]. Specifically multifocal disease, defined as >1 non-contiguous tumor, is associated with worse survival and more aggressive disease. Although there is no definition for sarcomatosis, a single institution series of 79 patients with multifocal RPS by Anaya et al. found a two-fold increased risk of death with >7 non-contiguous tumors [41].

The timing of potential re-resection is nuanced and individualized. As mentioned above, a long disease-free interval with a limited recurrent tumor burden which would allow for resection with an acceptable morbidity profile in a medically fit patient is viewed favorably. Tumor histology and response to systemic or locoregional therapy, as well as the option for clinical trial enrollment, may also be factored into the decision to submit a patient to repeat resection. Lastly, symptom palliation alone may warrant operative intervention. In patients with a short disease-free interval, a test of time with systemic chemotherapy may be warranted prior to rapidly returning to the operating theater. In the specific instance of suspected recurrent WDLPS, these tumors may be observed for a period to determine the trajectory of their disease. Park et al. previously reported in a series

of 105 patients with recurrent RPS that only tumors growing < 0.9 cm/month benefited from repeat resection. [43]. These tumors may be observed radiographically as a fat-dense mass; however, any solid or dense component should be biopsied to determine concurrent dedifferentiated component. Unfortunately, the decision to pursue resection of recurrent disease, as well as timing of surgery, cannot be distilled into an algorithm and requires both multi-disciplinary evaluation and informed discussions with the patient.

## 7. Conclusions

- Recurrence is common after surgery for RPS, and pattern of recurrence (local recurrence vs. distant metastasis) varies by histology.
- Patients with primary and recurrent RPS should be referred for multidisciplinary evaluation at a sarcoma center.
- High-quality, contrast-enhanced CT imaging of the chest, abdomen, and pelvis should be performed when tumor recurrence is first identified.
- MRI may be useful to help delineate pelvic disease or extent of tumor involvement.
- Neoadjuvant therapy may have some limited utility in downsizing large or locally invasive tumors, although data are limited.
- Data on the use of adjuvant therapy and radiation are limited in the setting of recurrent disease.
- Repeat resection is worth considering for technically resectable local and distant recurrence and has an associated survival benefit.
- In general, multifocal recurrence is associated with worse survival and should be reserved for palliation in symptomatic patients.

**Author Contributions:** Conceptualization, resources, data curation, J.S.J., C.P.R. and M.F.; writing—original draft preparation, J.S.J.; writing—review/revision, J.S.J., C.P.R. and M.F. All authors have read and agreed to the published version of the manuscript.

**Funding:** This research received no external funding.

**Informed Consent Statement:** Informed consent was obtained for patient images used in the manuscript. There were otherwise no experimental interventions in this review manuscript.

**Conflicts of Interest:** J.S.J., M.F., C.P.R. have no real or apparent conflict of interest to disclose.

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
