# Peer review of "Management of Recurrent Retroperitoneal Sarcoma"

_curroncol, doi:10.3390/curroncol30030209_

Round 1

Reviewer 1 Report

Thank you for the opportunity to review this manuscript.

This review is an excellent overview of all matters regarding the management of recurrence of retroperitoneal sarcoma. It is a very complete overview and I it can definitively be published in its current form.

One thing I would like to suggest is to add recommendations about timing of the "next" surgery. Is it wise to wait as long as possible with the next laparotomy? What are arguments not to postpone surgery? So to give some practical advices on decision-making in these cases. 

Author Response

We thank the reviewer for taking the time to evaluate and provide feedback on this manuscript. In brief, for the medically fit patient with a long disease-free interval and a resectable tumor burden, there is no benefit to waiting additional time. For patients with a short disease-free interval, a time-test of systemic therapy may be warranted. We have added substantial language to the "Patient Selection" section of the manuscript.

Reviewer 2 Report

This is a well written manuscript that summarizes the current litterature on the management of recurrent retroperitoneal sarcomas. Despite the shortcomings of the evidence at hand, I would recommend that the authors propose an algorithm on which patients should be offered salvage surgery at the time of recurrence based on the current evidence they highlighted throughout the manuscript.

Author Response

We thank the reviewer for taking the time to review this manuscript. Unfortunately, these tumors are too rare and the decision to pursue repeat resection is too nuanced to publish a true algorithm, however we have written additional language in "Patient Selection." Notably, in a medically fit patient with a resectable tumor burden that would allow for resection with an acceptable morbidity profile, we offer repeat resection. That being said, we want to heavily emphasis the importance of multi-disciplinary evaluation and consideration of histology, tumor burden, disease-free interval, response to other therapies, and patient fitness/comorbidities.

Reviewer 3 Report

The present paper is good and interesting.

The written English is clear. The scientific and clinical impact is acceptable.

However, in my opinion a minor revision is required before the publication.

Specifically, I suggest the following revisions:

-        abstract section:  the phrase “e.g., well-differentiated liposarcoma is more likely to recur locoregionally, whereas leiomyosarcoma is more likely to develop distant metastases” should be placed in parentheses.

-        introduction section: as a rule, the introduction should be more concise. There are too many concepts, typical related to a discussion section; authors write about of specific references, specific data and related results. They write “in a gronchi et al., …..” and “an analysis performed by Toulmonde et al.” or  “Bonvalot et al. analyzed…”…data have to be translate into concise concepts. A detailed analysis of results available into the current literature should be reserved for a specific section with a specific title such as “pattern of recurrences” ..or as the authors wish!

-        There is a specific recommendation about the use of the colon after each single subtitle? If not, it would be better to use a full stop instead of colon.

Author Response

We thank the reviewer for taking the time to evaluate and provide feedback on this manuscript. We have added the parenthetical to the abstract, as the reviewer indicated. We agree that there is an abundance of information in the introduction, however for a review article, we feel that this detail is required to appropriately discuss the topic at hand and propose recommendations. Regarding the colons after the subtitle, we are deferring to the MDPI formatting (which appears to use colons after subtitles).

Reviewer 4 Report

The submitted article discusses an evidence based review of the management of the difficult clinical scenario of a recurrent retroperitoneal sarcoma. The presented review article takes a multidisciplinary view spanning from diagnostics (radiology/pathology) to therapeutics (chemotherapy/radiation/surgery). 

This article is well written, comprehensive, and warrants publication with minor comments.

1. Can you discuss the role of an initial observation strategy, especially for WD liposarcoma, as a means of testing the biology of the tumor. (PMID: 19953716)

2. Chandrajit Raut's name was misspelled when submitting the metadata for the publication. It is correct in the manuscript. 

Author Response

We thank the reviewer for taking the time to read and critique our manuscript. We added language regarding observing WDLPS to the "Patient Selection" section of the manuscript. "Park and colleagues previously reported in a series of 105 patients with recurrent RPS that only tumors growing < 0.9cm/month benefited from repeat resection. [44] These tumors may be observed radiographically as a fat-dense mass, however any solid or dense component should be biopsied to determine concurrent dedifferentiated component."

Thank you for pointing out the discrepancy and misspelling of Dr. Raut's name, we will bring this to the attention of MDPI and the editors. 

Reviewer 5 Report

Dr. Jolissaint and co-authors provide an extremely well-written and comprehensive review on the management of recurrent retroperitoneal sarcoma.

Comments:

1. The authors appropriately describe the varied histologic subtypes, differences in behaviors, and locoregional vs. distant metastatic potential.

2. The need for care in a high-volume centers with pathologic expertise is emphasized.

3. The controversial role of chemotherapy is appropriately referenced. While EORTC 62931 is highlighted, it would be worth mentioning that the majority had extremity tumors and that RP tumors were underrepresented.

4. The manuscript would also be strengthened by mentioning/describing the ongoing STRASS 2 study given its potential to be practice changing for patients with RP disease (albeit not specifically for recurrent RP disease).

Author Response

We thank the reviewer for taking the time to evaluate and provide feedback on this manuscript. Regarding points #3 and 4:

3. We have added language regarding the low proportion of patients with RPS/high proportion with extremity sarcoma: "It is important to note that only 21% of this study cohort had a central disease burden, with the remainder being either extremity or limb girdle." 

4. We have added language regarding the ongoing STRASS 2 trial: "Although limited to primary DDLPS and LMS, the STRASS 2 Trial (EORTC 1809, NCT04031677) is currently enrolling patients to determine whether neoadjuvant chemotherapy improves outcomes, as measured by DFS, OS, and RFS."